



# LUCAS Copernicus 2018: Earth Observation relevant in-situ data on land cover throughout the European Union

Raphaël d'Andrimont[1], Astrid Verhegghen[1], Michele Meroni[1], Guido Lemoine[1], Peter Strobl[1], Beatrice Eiselt[2], Momchil Yordanov[1], Laura Martinez-Sanchez[1], and Marijn van der Velde[1]

[1]European Commission Joint Research Centre (JRC), Ispra, Italy
[2]European Commission, Eurostat (ESTAT), Luxembourg, Luxembourg

**Correspondence:** Raphaël d'Andrimont (raphael.dandrimont@ec.europa.eu) and Marijn van der Velde (marijn.van-der-velde@ec.europa.eu)

**Abstract.**

The Land Use/Cover Area frame Survey (LUCAS) is a regular in-situ land cover and land use ground survey exercise that extends over the whole of the European Union. LUCAS was carried out in 2006, 2009, 2012, 2015, and 2018. A new LUCAS module specifically tailored to Earth Observation was introduced in 2018: the LUCAS Copernicus module, aiming

at surveying land cover extent up to 51 meters in four cardinal directions around a point of observation. This paper first summarizes the LUCAS Copernicus protocol to collect homogeneous land cover on a surface area of up to a 0.52 ha. Secondly, it proposes a methodology to create a ready-to-use dataset for Earth Observation land cover and land use applications with high resolution satellite imagery. As a result, a total of 63 364 LUCAS points distributed over 26 level-2 land cover classes were surveyed on the ground. Using homogeneous extent information in the four cardinal direction, a polygon was delineated for

each of such point. Through geo-spatial analysis and by semantically linking the LUCAS core and Copernicus land cover observations, 58 428 polygons are provided with a level-3 land cover (66 specific classes including crop type) and land use (38 classes) information as inherited from the LUCAS core observation. The open-access dataset supplied with this manuscript (https://doi.org/10.6084/m9.figshare.12382667.v3 (d'Andrimont, 2020)) provides a unique opportunity to train and validate decametric sensor-based products such as those obtained from the Copernicus Sentinel-1 and -2 satellites. A follow-up of the

LUCAS Copernicus module is already planned for 2022. In 2022, a simplified version of the LUCAS Copernicus module will be carried out on 150 000 LUCAS points for which in-situ surveying is planned. This guarantees a continuity in the effort to find synergies between statistical in-situ surveying and the need to collect in-situ data relevant for Earth Observation in the European Union.

## 1 Introduction

The Land Use/Cover Area frame Survey (LUCAS) is a regular in-situ land cover and land use data collection exercise that extends over the whole of the European Union (EU) (Gallego and Delincé, 2010; Eurostat, 2018c). LUCAS has been carried out in 2006, 2009, 2012, 2015, and 2018. During these five campaigns, a total of 1 351 293 points were surveyed and 5.4 million photos were collected. On each of these surveyed points, observations were recorded on up to 109 variables. The





combination of the information collected in the five LUCAS surveys has resulted in the most comprehensive in-situ database
on land cover and land use in the EU (d'Andrimont et al., 2020).

LUCAS in-situ data collection was designed for EU-wide standardized reporting of land cover and land use area statistics
and not for training and validation of remote sensing data algorithms. The LUCAS activity is complementary to the CORINE
Land Cover (CLC) inventory that collects land cover data by interpreting satellite images and orthophotos. In addition, in
2018 the Copernicus High Resolution Layers (HRL) have been produced to provide information about different land cover
characteristics. Five HRLs describe some of the main land cover characteristics: impervious (sealed) surfaces (e.g. roads and
built up areas), forest areas, grasslands, water and wetlands, and small woody features.

In the scientific community, LUCAS has been widely used for soil studies thanks to the topsoil survey module (Orgiazzi
et al., 2018). LUCAS data has also already been valuable in the context of land cover and land use research and remote sensing
specifically. Esch et al. (2014) used the data for crop type mapping in the North of Germany. Zillmann et al. (2014) provided
an accuracy assessment of grasslands mapping in Hungary based on LUCAS. Mack et al. (2017) used Landsat time series
along with LUCAS in-situ data to generate a land use and land cover product for Germany. Leinenkugel et al. (2019) assessed
the potential of open geodata including LUCAS to generate land use and land cover products from multi-temporal Landsat
satellite observations over 3 European sites. Pflugmacher et al. (2019) recently demonstrated the potential of using LUCAS
to map pan-European land cover (13 classes) with Landsat data. Close et al. (2018) provided a Sentinel-2 LUCAS-based
classification over Southern Belgium in the context of Land Use, Land Use Change, and Forestry (LULUCF) monitoring.
According to Weigand et al. (2020), LUCAS in-situ data is a suitable source for classifying high-resolution Sentinel-2 imagery
at a large scale. Weigand et al. (2020) tested the accuracy of different pre-processing approaches of the LUCAS data based on
positioning and semantic selection. These studies highlight that there is an interest and value to the remote sensing research
community in using LUCAS in-situ data. Nevertheless, the LUCAS core protocol has major limitations in terms of spatial
scale and representativeness when it comes to collecting in-situ data for calibration, training, and/or validation of EO products.

While LUCAS survey data had been valuable in providing in-situ observations relevant for remote sensing as highlighted,
the LUCAS survey was designed to collect statistics and thus has inherent shortcomings when used in the context of EO. In
2018, a new LUCAS module specifically tailored to Earth Observation (EO) was introduced: the LUCAS Copernicus module.
The Copernicus module was designed to improve the value of LUCAS in-situ surveying for EO and to address the three specific
EO limitations described hereafter.

First, the quality of EO derived products is underpinned by the availability and thematic representativity of precisely geo-
located in-situ observations. Such in-situ data is essential to train and validate algorithms applied to EO products. Comprehen-
sive and thematically rich in-situ data can lead to better classifiers and more accurate multi-temporal land surface mapping.

Second, remotely sensed observations of the Earth are increasingly frequent along with finer spatial and spectral detail and,
in the case of the observations by the fleet of Sentinel satellites of the EU's Copernicus Earth Observation Program, accessible
to everyone. These remote observations need complementary in-situ observations. At the same time, there is an enormous and
continuing growth in a variety of services relying on geo-location. In this context, it is fair to say that we are witnessing a
renewed recognition of the importance of in-situ data for EO.





Therefore, the third motivation is that the LUCAS Copernicus collected in-situ data should be representative, comprehensive, precisely geo-located, available over larger areas, available across political borders, and with open access. Free and open accessibility is in fact essential for contributing to the creation of common in-situ data-sets and protocols as currently pursued by e.g. the Land Product Validation (LPV) of the Working Group on Calibration and Validation of the Committee on Earth Observation Satellites (CEOS) and by the Joint Experiment for Crop Assessment and Monitoring (JECAM). The availability of such data-set acquired with transparent protocols is key to assess the quality EO products resulting from various public and commercial activities. Thus, the Copernicus module gives the opportunity to further integrate the classical LUCAS survey purpose of collecting statistically representative information with the need to collect in-situ data to produce better EO-derived products, specifically for the EU's Copernicus program. The Copernicus module equips the EU with an in-situ dataset specifically fitting EO land applications monitoring allowing to develop consistent land monitoring at EU level.

While available since 2019, the Copernicus module has not been used in EO applications (to the best of our knowledge). This study is reducing the complexity of the data to ease the take up of the Copernicus module by the remote sensing community.

This manuscript describes and provides the LUCAS Copernicus data in a ready-to-use format. More specifically, this study (i) describes the LUCAS 2018 Copernicus in-situ survey protocol, (ii) presents a methodology to produce polygons from the surveyed data to be used in EO studies, (iii) proposes a method to inherit more detailed information from the LUCAS core, and (iv) highlights the added value of the survey in order to derive a simplified protocol for the LUCAS Copernicus module that will be integrated in future LUCAS surveys (e.g. in 2022).

## 2 LUCAS 2018 data

In 2018, the campaign involved more than 1300 actors including more than 900 surveyors and lasted for 23 months. The actual in-situ data collection occurred between March and September 2018. The raw data have been available online since May 2019 (Eurostat, 2019a) as a downloadable csv table with 97 columns and 337 854 records (Table 4 presents the attribute names of the 97 original fields; a record descriptor is available in Eurostat (2019c); the detailed survey instructions in Eurostat (2018d)). At the basis of the LUCAS sampling methodology is the LUCAS grid containing the theoretical locations of the points. This LUCAS 2x2 km$^2$ grid (i.e. one point every 2 km in the EU) consists of geo-referenced points in the EU from which the sample is selected for each survey. The grid is static and includes 1 090 863 points stratified according to land cover class. The grid is available in csv format from Eurostat (2019b). For a detailed description of the grid data see Eurostat (2018a), and for technical details about the stratification see Eurostat (2018b). Out of the 337 854 points surveyed in 2018, 23% points had been included in three previous surveys (2009, 2012, and 2015), 25% had already been surveyed once or twice before (e.g. in 2009 and 2015), and the remaining 52% of the points were new entries. In the LUCAS 2018 survey, 70.45% of the points were surveyed in-situ, and 29.54% were obtained through the interpretation of detailed ortho-photos (Table 1 and Table A1).

Eurostat has carried out LUCAS surveys every three years with the survey design ever evolving, however the LUCAS *core* component (i.e. the identification of the point, and the surveying of specific variables on different aspects of land cover, land use, and land and water management Eurostat (2018d)), has remained comparable for all five surveys. At each LUCAS point,



standard variables are collected including land cover, land use, environmental parameters, and landscape photos. Additionally to the *core* variables collected, other specific *modules* were carried out on demand such as (i) the transect of 250 m to assess transitions of land cover and existing linear features (2009, 2012, 2015), (ii) the topsoil module (2009, 2012 (partly), 2015 and
2018), (iii) the grassland (2018), and (iv) the Copernicus module (2018).

Out of the 337 854 LUCAS points sampled in 2018 (combining in-situ and photo-interpreted points, Table A1), the Copernicus module was programmed for 90 620 points and actually executed for 63 364 points (Table 1). For 27 256 (30.08%) planned points, the surveyors did not manage to reach the point to make the observation, for example due to natural or human-made obstacles. Therefore, the Copernicus module was carried out in-situ for a total of 63 364 points, corresponding thus to 69.02%
of the planned Copernicus points.

**Table 1.** LUCAS 2018 points totaling 337 854 points. The points could be either In-situ (238 014, 70.45%), Office Photo-Interpreted (99 803, 29.54%) or Others (i.e. "In situ PI not possible", "Out of national territory" or "Out of EU28"). Within the in-situ class, the observation was done on the point following the LUCAS core protocol and for a subset of locations also for the Copernicus module.

|  | LUCAS Core Points | LUCAS Core and Copernicus | Total |
| --- | --- | --- | --- |
| **In situ** | 147 394 + 27 256* | 63 364 | 238 014 |
| **Office Photo-Interpreted** | 99 803 | - | 99 803 |
| **Others** | 37 | - | 37 |
| **Total** | 274 490 | 63 364 | 337 854 |

*\* Planned in the Copernicus module, but the Copernicus survey was not possible and thus solely surveyed as a point*

## 3   LUCAS 2018 Copernicus Protocol

In the LUCAS core protocol, the surveyor aims to get as close as possible to the theoretical point. The surveyor then provides the so-called LUCAS *core* observations for the LUCAS theoretical point from the location that the surveyor was actually able to reach. Thus, although typically close to each other, the nominal geolocation of a LUCAS point may not exactly coincide with
the actual observation location, that is not recorded for LUCAS core points. As an illustrative example, the observation is made from an unknown location and assigned to the LUCAS nominal point in red on Figure 1. The exact geolocation of the surveyor observation is recorded only in the corresponding LUCAS-Copernicus entry (green point in Figure 1). The LUCAS theoretical grid point observation is representative for a circle of 1.5 meter radius. In some specific cases, the window of observation is extended to a 20 meter radius whenever the land cover at the point is heterogeneous (Eurostat, 2018d). This occurs in areas
such as permanent crops (B7X, B8X, except nurseries B83) where the parcels of permanent crops contain trees or other plants along with with bare soils and/or grassland or another crop, in woodland (CXX), shrub land (DXX) where a mix of e.g. shrubs and trees might occur, in grassland (EXX) where land features may alternate (e.g. grassland with trees), in bare land (FXX)



and in wetland (HXX). Given the mentioned protocol, two main drawbacks of the LUCAS core observations are apparent for their use in the context of EO applications.

The first limitation is that the observation corresponds to a subfraction of a Sentinel(-1 and -2) pixel (the circle with 1.5 m radius, representing thus an area of $7.07\,\mathrm{m}^2$) and is thus not directly usable with such decametric sensors. Indeed, the 10-m pixel (i.e. $100\,\mathrm{m}^2$) could be covered by different land covers while the LUCAS observation only captures one. This jeopardizes the use of LUCAS core observations for training and validation when building EO derived products. The second limitation refers to GPS geo-location survey inaccuracies that is comparable to the representative area, making the information unsuitable. On

the contrary to address these limitations, LUCAS Copernicus module collected the exact geolocation of the observation as well as information on the spatial extent and homogeneous continuity of the Land Cover (LC) observed around the point, making it suitable for use in EO applications.

More specifically, the following additional data are collected on the LUCAS Copernicus surveyed points: (i) the exact location of the observation, and (ii) the land cover (level 2) extent up to 51 m from the point in the four cardinal directions (N,

W, S, E), as well as the neighboring LC. Note that the surveyor records 51 m to indicate that the land cover is homogeneous for more than 50 m. However, as the exact extent is not reported, we conservatively set it to the minimum extent of 51 m. Figure 1 illustrates the Copernicus protocol for one point and the respective collected data is shown in Table 2. On the basis of these LUCAS Copernicus observations, a quadrilateral polygon with homogeneous LC can be constructed. As part of the Copernicus module, the surveyor collects 13 additional variables and three types of observations (Table 2): the level-2 LC (one

variable); the extent of the Copernicus land cover (LUCAS LC classification at level 2) registered at the point reached in the field (four variables); the next land cover (up to 50 m) (four variables) and the breadth of the next land cover (four variables). The breadth corresponds to the % of the width of the land cover in this sector, as visible on the landscape photo. This means that the breadth is 100% if the next LC is seen all over the photo from one side to the other. If the next land cover is not visible on the photo because it is completely behind a linear feature (e.g. hedge) or because it is completely hidden by the terrain, then

the next land cover is to be recorded but the breadth is 0%. For more information about the breadth and the next land cover, see Eurostat (2018d).

The following sections describe how the LUCAS Copernicus data are prepared and cleaned to obtain a ready to use data set provided with this manuscript. The following workflow was done in R (Code and Data availability, see Section 8).

## 3.1 Adding an explicit LUCAS land cover and land use legend

The LUCAS land cover classification is hierarchical and contains four levels briefly described hereafter (for a detailed description, see the Technical reference document C3 Classification (Eurostat, 2018e)). The land cover classification system is subdivided into eight main level-1 land cover categories: Artificial Land, Cropland, Woodland, Shrubland, Grassland, Bare Land, Water and Wetlands. The legend level-2 contains 26 classes (e.g. 8 under level-1 B Cropland) and level-3 comprises 73 classes (e.g. 9 under level-2 B1 Cereals). Only a limited number of observations has a level-4 land cover information distributed

into 205 classes ("LC1_SPEC" field in the data). Similarly, the Land Use comprises 40 subclasses.



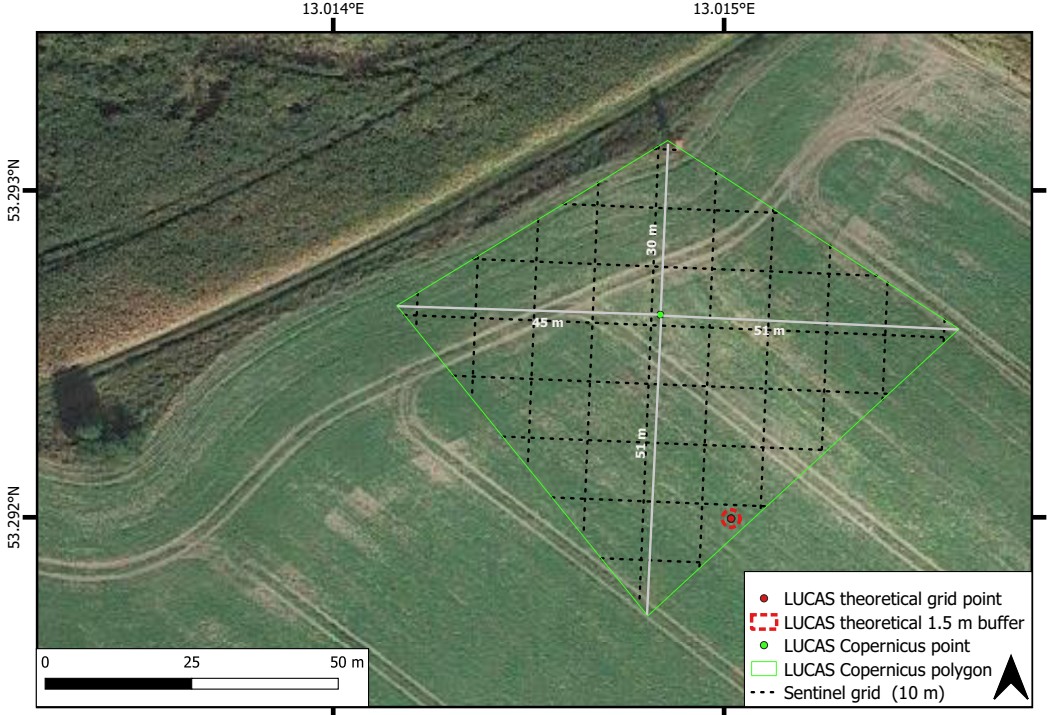

**Figure 1.** Building the Copernicus polygon geometry (example for POINT ID 45223358). The collected Copernicus variables (Table 2) are used to build the geometry of the Copernicus polygon. As the LUCAS theoretical point is inside in the Copernicus polygon, the LC legend of the LUCAS theoretical observation (here B32 - Rape and turnip rape) could be inherited to the Copernicus polygon (B3 - Non-permanent industrial crops) as described in Section 4. The background RGB imagery is obtained from "Map data ©2019 Google".

To facilitate the usability of the data, in addition to the code describing the land use or land cover (e.g. B21 or U112), an explicit legend label was added to the dataset provided with this manuscript. This was done by adding nine Label explicit fields (Table 4) to the data for the LC and LU legend.

In the results section, details on the hierarchical legend structure classes are also provided (Table 3 on legend level-2, Figure
5 on legend level-3 ("LC1" field in the data), and the 40 Land Use sub classes as shown by the level-3 distribution of the Copernicus polygons in Table A2), "LU1" in the data).

### 3.2 Constructing the LUCAS Copernicus polygon

On the basis of the LUCAS Copernicus observations, a polygon with homogeneous LC can be constructed. In order to generate the LUCAS Copernicus polygon, the Copernicus point (i.e the effective location of the surveyor, i.e. "GPS_LAT" and
"GPS_LONG") is defined as the centre to build a quadrilateral for each point. The location of this point is first projected in the Lambert Azimuthal Equal-Area projection coordinate reference system (ETRS89-LAEA). Second, the 4 distances (N, W,





**Table 2.** Example of information collected by the Copernicus protocol (for point with ID 45223358). The Copernicus protocol collects observations on 13 variables: land cover (LC) at LUCAS legend level 2 (here B3 is "Non-permanent industrial crops"), the extension of the LC in the four cardinal directions (up to 50 m, 51 means more than 50 m), the breadth of the next LC in the four cardinal directions (%) and the next LC in the four directions (here, E2 means "Grassland without tree/shrub cover" in the N and W). Figure 1 shows how this information is used to build the geometries of the Copernicus polygon with homogeneous LC. The radial distance "d" is measured between the Copernicus point and the next LC, with "888" and "8" meaning "not relevant".

| | **Copernicus LC: B3** | | |
|---|---|---|---|
| | **Extension (d) of Copernicus LC in cardinal directions in m** | **Breadth of next LC in (%), if d <= 50m** | **Next LC, if d <= 50m** |
| **N** | *30* | *15* | *E2* |
| **E** | *51* | *888* | *8* |
| **S** | *51* | *888* | *8* |
| **W** | *45* | *50* | *E2* |

S, E) measured by the surveyor are added to the point in the four respective cardinal directions resulting in 63 364 irregular quadrilaterals. The quadrilateral diagonals can measure up to 102 m, but are smaller if the surveyor found a field boundary within 51 m of the LUCAS-Copernicus point.

## 3.3 Quality check

While the Copernicus protocol was implementable for 63 364 points, several surveyed point locations (i.e. LUCAS Copernicus point as defined by "GPS_LAT" and "GPS_LONG") were either missing or wrong. The missing locations could be flagged for 67 points ("GPS_PREC"=8888 or "GPS_LAT"=0 or "GPS_LONG"=0 ). In addition to these, 10 points were discarded because the surveyor geolocation ("GPS_LAT", "GPS_LONG") was far away from the nominal location ("TH_LAT" , "TH_LONG") ), i.e. difference larger than 0.1 degree (i.e. about 11.1 km). In addition to the missing GPS measured locations, some macro errors were flagged and removed by selecting points for which the longitude and latitude differences between the GPS measured location and theoretical location ("TH_LAT" , "TH_LONG") is larger than 0.1 degree. This allows to flag and remove 10 polygons which are all wrongly located because of the "GPS_EW" field (i.e. GPS Observation East/West). This location quality check permits to flag and remove a total of 75 polygons resulting in a final total of 63 287 polygons.

## 3.4 Resulting LUCAS Copernicus Data

The 63 287 Copernicus polygons surveyed are published along with this paper. They are distributed among 26 level-2 LC classes (Table 3) in eight level-1 LC classes (see map in Figure 3).

The homogeneous area of the 63 287 polygons range between 0.005 ha to 0.52 ha with an average of 0.32 ha (Figure 2) corresponding to 32 10-m pixels. Half of the polygons are larger than 0.33 ha. Also, the third quartile corresponds to the



maximum area of 0.52 ha, which is the maximum area possible for a rhombus with diagonals of 102 m (51 m + 51 m). Among the 63 287, it is worth mentioning that 21 657 polygons (i.e. 34.2%) have an area greater than 0.5 Ha, i.e. corresponding thus to more than 50 10-m pixels. These characteristics make the obtained spatial data well suited for training and validation of products based on decametric (i.e. 0,01 ha) and even subdecametric remote sensing sensors.

**Table 3.** Distribution of level-2 land cover classes of the resulting LUCAS Copernicus polygons (N=63 287).

| Class | Label | # LUCAS Copernicus polygons |
|---|---|---|
| Roofed build-up areas | A1 | 12 |
| Artificial non-built up areas | A2 | 374 |
| Other artificial areas | A3 | 21 |
| Cereals | B1 | 12 774 |
| Root crops | B2 | 877 |
| Non-permanent industrial crops | B3 | 2435 |
| Dry pulses, vegetables and flowers | B4 | 767 |
| Fodder crops | B5 | 2757 |
| Permanent crops: fruit trees | B7 | 817 |
| Other permanent crops | B8 | 1148 |
| Broadleaved woodland | C1 | 8481 |
| Coniferous woodland | C2 | 5996 |
| Mixed woodland | C3 | 4484 |
| Shrubland with sparse tree cover | D1 | 1308 |
| Shrubland without tree cover | D2 | 1546 |
| Grassland with sparse tree/shrub cover | E1 | 2078 |
| Grassland without tree/shrub cover | E2 | 13 053 |
| Spontaneously re-vegetated surfaces | E3 | 2608 |
| Rocks and stones | F1 | 35 |
| Sand | F2 | 30 |
| Lichens and moss | F3 | 3 |
| Other bare soil | F4 | 1503 |
| Inland water bodies | G1 | 1 |
| Inland running water | G2 | 9 |
| Inland wetlands | H1 | 164 |
| Coastal wetlands | H2 | 6 |
| | TOTAL | 63 287 |

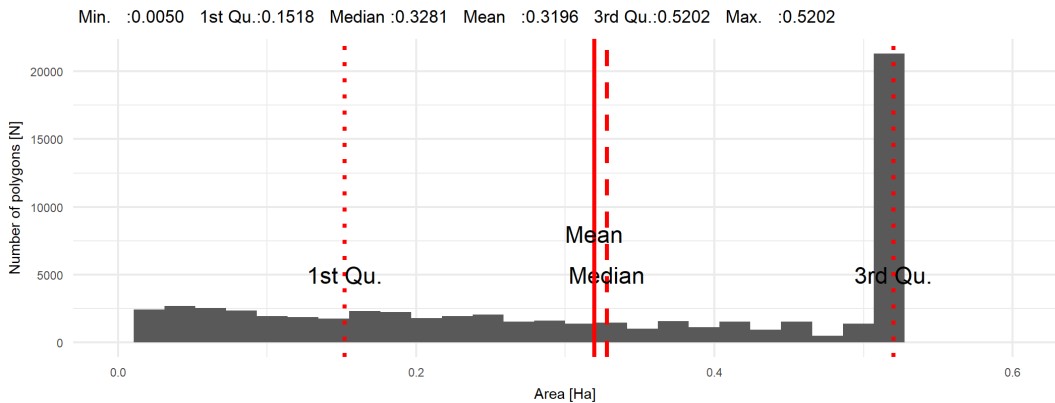

**Figure 2.** Distribution of the area of the LUCAS Copernicus polygons in Ha (N=63 287). On average, the polygon covers an area of 0.32 Ha.

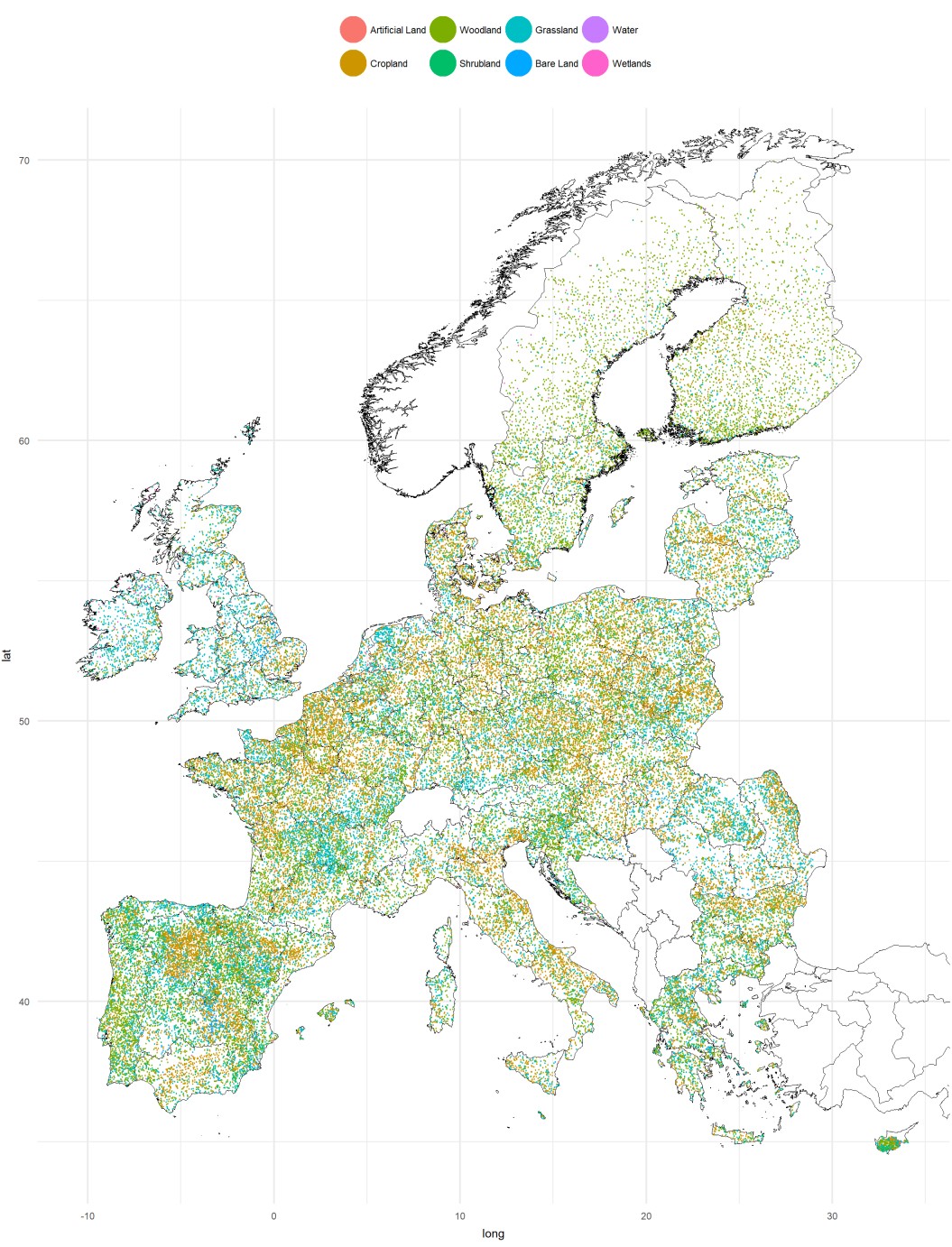

**Figure 3.** Map of LUCAS Copernicus polygons (N=63 287) surveyed in 2018 per level-1 land cover class over the EU28.





## 4   Linking LUCAS core data to Copernicus polygons

A set of rules was defined to link LUCAS core and LUCAS Copernicus data and thus enrich the LUCAS Copernicus set of information. The rationale is that if the LUCAS theoretical point location falls within the LUCAS polygon, the LUCAS core surveyed attributes at the theoretical point could be inherited to the LUCAS Copernicus polygon. This condition is satisfied by the vast majority of the polygons (60 134 points, i.e. 95.02%). In addition, to filter out suspicious data points where the LUCAS core and Copernicus information were not in agreement despite being spatially consistent, we retained only those points where

the reported Copernicus level-2 land cover observed is the same as the one reported for the LUCAS theoretical point (50 417 points, i.e. 95.47%). This happens when a surveyor can observe the LUCAS theoretical point from a distance, but makes the Copernicus observation on the actual point that was reached. On this Copernicus point, the land cover does not correspond to the land cover of the LUCAS theoretical point. Among the 63 287 Copernicus polygons available with this paper, 58 428 polygons (i.e. 92.23%) fulfill both requirements (Condition "CPRN_LC_SAME_LC1" and condition "LUCAS_CORE_INTERSECT"

in the provided dataset) and are thus flagged as "COPERNICUS_CLEANED" in the data. For these polygons, the more detailed level-3 land cover class of LUCAS core can be inherited to the LUCAS Copernicus polygon ("COPERNICUS_CLEANED" is "TRUE"). Figure 4 illustrates the variety in shapes of the constructed quadrilateral Copernicus polygons as projected on top of satellite imagery for different land cover types. The resulting polygons are distributed over 66 specific LC classes as shown in Figure 5. Similarly, the level-3 Land Use (LU) is also available distributed in 38 classes organised in four main classes (see

Table A2).



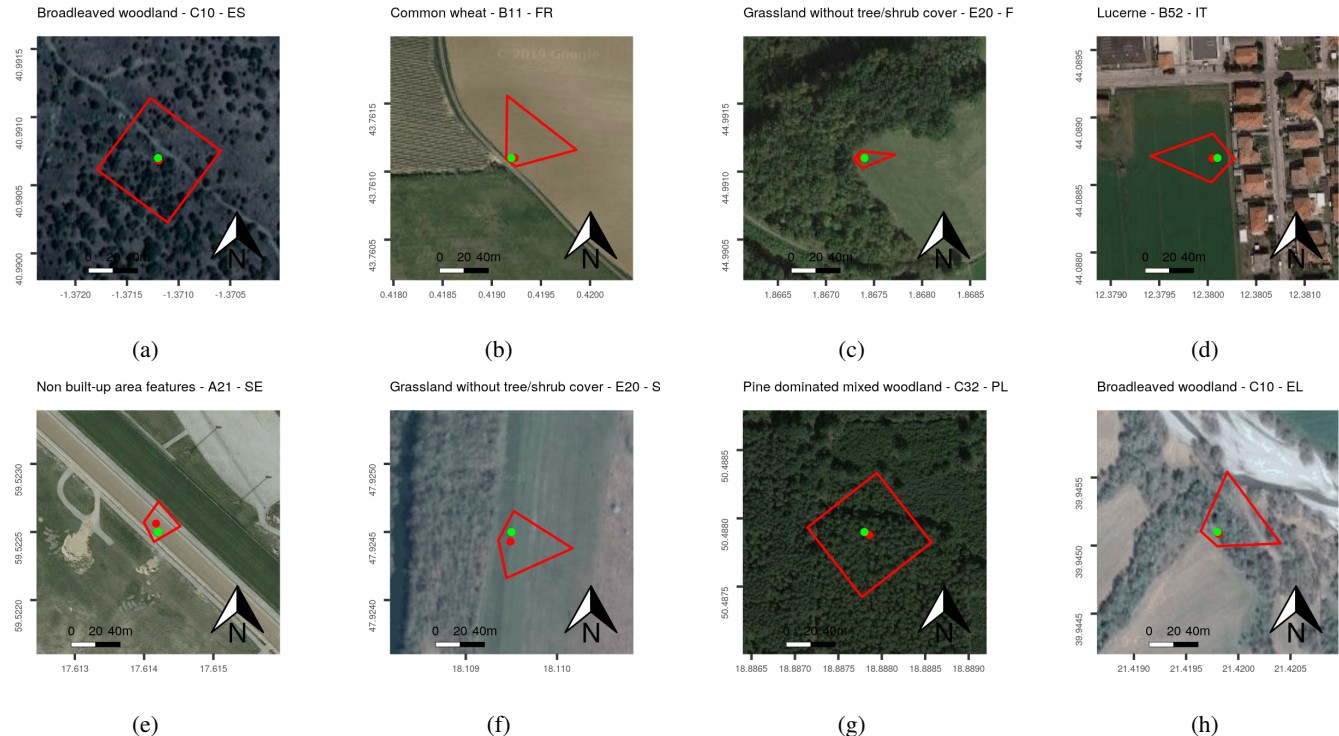

**Figure 4.** Examples of LUCAS Copernicus built polygons. The green point is the theoretical LUCAS point. The red point is the GPS location of the Copernicus surveyor. Polygons are built using distances in N, E, S, W directions collected on the ground. The background RGB imagery is obtained from "Map data ©2019 Google".



**Figure 5.** Distribution of level-3 land cover (inherited from LUCAS core) for the LUCAS Copernicus polygons (N=58 428).





## 5   Public data and usage note

## 6   Discussion

The LUCAS Copernicus polygons and data compiled and presented here can provide valuable information for a variety of top-
ics and applications. The LUCAS Copernicus polygons can provide valuable information to extract land cover specific surface
radiometric and temporal signatures as measured by different sensors, in the multispectral, thermal and microwave range, for
different land cover types. This is particularly relevant for land covers exhibiting a dynamic signal (e.g. forests, grasslands,
crops) that is modulated by climatic and agro-ecologic conditions, which are well sampled in this EU-wide dataset. The dataset
can serve various EO-based applications, among many others: train classification algorithms for land cover mapping using
existing sensors (e.g. Sentinel 1 and 2), validate land cover products centred on 2018 (e.g. the Copernicus High Resolution
Layers), study land cover specific land surface processes (e.g. phenology), develop algorithms to monitor crop and grassland
management practices.

With this paper we provide LUCAS Copernicus polygons constructed at 63 287 locations. In addition, we provide a dataset
that benefits from inheriting attributes collected on those same points via the LUCAS core protocol. This results in 58 428
Copernicus polygons, discarding a total of 4859 polygons.

The LUCAS Copernicus module is also planned to be carried out during the LUCAS 2022 survey. However, a simplified
protocol has been designed for the LUCAS 2022 survey. In this protocol, the observations on the distance of homogeneous
LC from the point, and the LC remain, but observations on the neighboring LC and breadth of the neighboring LC have
been discarded. Despite this simplification, the coverage of LUCAS Copernicus module will be expanded in 2022 to 150 000
LUCAS points for which in-situ surveying is planned.

## 7   Conclusions

For the first time, the LUCAS 2018 survey contained a module that was specifically tailored to the needs of EO. The LUCAS
Copernicus module collected homogeneous land cover data over areas with a size relevant to 10-m satellite sensors. A total of
63 364 Copernicus polygons were obtained across the EU representing 66 land cover type classes at LUCAS legend level-2.
A follow-up of the LUCAS Copernicus module is planned for 2022. In 2022, a simplified version of the LUCAS Copernicus
module will be carried out on 150 000 LUCAS points for which in-situ surveying is planned. This guarantees a continuity in the
effort to find synergies between statistical in-situ surveying and the need to collect in-situ data relevant for Earth Observation
in the European Union.

## 8   Code and data availability

The data repository https://doi.org/10.6084/m9.figshare.12382667.v3 (d'Andrimont, 2020) contains the following files:

– *LUCAS_2018_Copernicus_polygons.shp* : Shapefile of polygons with the "POINT_ID" attribute





- – *LUCAS_2018_Copernicus_attributes.csv* : CSV file containing the 109 variables including the "POINT_ID"

- – *CreateCopernicusPolygonsAll.R* : The open-source R script used to generate the data

- – *LUCAS_2018_Copernicus_ReadMe.txt* : short description of the data repository

The LUCAS Copernicus 2018 dataset is provided as a polygon shapefile along with a csv table containing 109 attributes and

available. Among the 109 attributes (list in Table 4), 97 attributes are the original fields as described in Eurostat (2019c), nine attributes are the legend-explicit LC and LU obtained as described in Section 3.1 and three attributes are obtained as described in the previous Section 4.

To use the data, the attribute "POINT_ID" should be used to join the attribute table of the shapefile and the csv table. While the Copernicus related level-2 LC could be used for every polygon, the level-3 LC and LU, along with other LUCAS core

information, should be used only for polygons with "COPERNICUS_CLEANED" as "TRUE" as described in the previous section.

**Table 4.** The LUCAS Copernicus 2018 dataset is provided as a polygon shapefile along with a table with 109 attributes to be joined based on POINT_ID. Among the 109 attributes, 97 attributes are the original as described in Eurostat (2019c), 9 are the legend-explicit LC and LU attribute obtained as described in d'Andrimont et al. (2020) and 3 attributes are obtained as described in Section 8.

| Origin of attributes | # | Attribute names |
|---|---|---|
| Original fields | 97 | POINT_ID, NUTS0, NUTS1, NUTS2, NUTS3, TH_LAT, TH_LONG, OFFICE_PI, EX_ANTE, SURVEY_DATE, CAR_LATITUDE, CAR_EW, CAR_LONGITUDE, GPS_PROJ, GPS_PREC, GPS_ALTITUDE, GPS_LAT, GPS_EW, GPS_LONG, OBS_DIST, OBS_DIRECT, OBS_TYPE, LC1, LC1_SPEC, LC1_PERC, LC2, LC2_SPEC, LC2_PERC, LU1, LU1_TYPE, LU1_PERC, LU2, LU2_TYPE, LU2_PERC, PARCEL_AREA_HA, TREE_HEIGHT_SURVEY, TREE_HEIGHT_MATURITY, FEATURE_WIDTH, LNDMNG_PLOUGH, LM_PLOUGH_SLOPE, LM_PLOUGH_DIRECT, LM_STONE_WALLS, CROP_RESIDUES, LM_GRASS_MARGINS, GRAZING, SPECIAL_STATUS, LC_LU_SPECIAL_REMARK, CPRN_CANDO, CPRN_LC, CPRN_LC1N, CPRNC_LC1E, CPRNC_LC1S, CPRNC_LC1W, CPRN_LC1N_BRDTH, CPRN_LC1E_BRDTH, CPRN_LC1S_BRDTH, CPRN_LC1W_BRDTH, CPRN_LC1N_NEXT, CPRN_LC1E_NEXT, CPRN_LC1S_NEXT, CPRN_LC1W_NEXT, CPRN_URBAN, CPRN_IMPERVIOUS_PERC, INSPIRE_PLCC1, INSPIRE_PLCC2, INSPIRE_PLCC3, INSPIRE_PLCC4, INSPIRE_PLCC5, INSPIRE_PLCC6, INSPIRE_PLCC7, INSPIRE_PLCC8, EUNIS_COMPLEX, GRASSLAND_SAMPLE, GRASS_CANDO, WM, WM_SOURCE, WM_TYPE, WM_DELIVERY, EROSION_CANDO, SOIL_STONES_PERC, BIO_SAMPLE, SOIL_BIO_TAKEN, BULK0_10_SAMPLE, SOIL_BLK_0_10_TAKEN, BULK10_20_SAMPLE, SOIL_BLK_10_20_TAKEN, BULK20_30_SAMPLE, SOIL_BLK_20_30_TAKEN, STANDARD_SAMPLE, SOIL_STD_TAKEN, ORGANIC_SAMPLE, SOIL_ORG_DEPTH_CANDO, PHOTO_POINT, PHOTO_NORTH, PHOTO_EAST, PHOTO_SOUTH, PHOTO_WEST |
| Label explicit fields | 9 | LC1_LABEL, LC2_LABEL, LC1_SPEC_LABEL, LC2_SPEC_LABEL, LU1_LABEL, LU2_LABEL, LU1_TYPE_LABEL, LU2_TYPE_LABEL, CPRN_LC_LABEL |
| EO application fields | 3 | CPRN_LC_SAME_LC1, LUCAS_CORE_INTERSECT, COPERNICUS_CLEANED |





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





**Table A1.** Copernicus survey with relation to observation type (OBS_TYPE), observation direction (OBS_DIRECT) and parcel area (PARCEL_AREA_HA).

| | | CPRN_CANDO | | | |
| --- | --- | --- | --- | --- | --- |
| | | Yes | No | Not Relevant | TOTAL |
| **OBS_TYPE** | | | | | |
| 1 | In situ < 100 mt | 63364 | 18924 | 121673 | 203961 |
| 2 | In situ > 100 mt | 0 | 2488 | 8671 | 11159 |
| 3 | In situ PI | 0 | 5844 | 17050 | 22894 |
| 4 | In situ PI not possible | 0 | 2 | 23 | 25 |
| 5 | Out of national territory | 0 | 0 | 10 | 10 |
| 6 | Out of EU28 | 0 | 0 | 2 | 2 |
| 7 | In Office PI | 0 | 0 | 99803 | 99803 |
| **OBS_DIRECT** | | | | | |
| 1 | On the point | 63057 | 23641 | 237089 | 323787 |
| 2 | Look to the North | 251 | 2834 | 8191 | 11276 |
| 3 | Look to the East | 56 | 781 | 1917 | 2754 |
| 8 | Not Relevant | 0 | 2 | 35 | 37 |
| **PARCEL_AREA_HA** | | | | | |
| 1 | Area < 0.1 | 855 | 3526 | 13965 | 18346 |
| 2 | $0.1 \le$ area $< 0.5$ | 2743 | 2876 | 12979 | 18598 |
| 3 | $0.5 \le$ area $< 1$ | 4168 | 2446 | 12917 | 19531 |
| 4 | $1 \le$ area $< 10$ | 25733 | 8434 | 70371 | 104538 |
| 5 | area $\ge 10$ | 29865 | 9975 | 136974 | 176814 |
| 8 | Not Relevant | 0 | 1 | 26 | 27 |
| | TOTAL | 63364 | 27258 | 247232 | 337854 |



**Table A2.** LUCAS Copernicus level-3 land use distribution.

| Class | # LUCAS Copernicus polygons |
|---|---|
| U111 - Agriculture (excluding fallow land and kitchen gardens) | 33 600 |
| U112 - Fallow land | 1781 |
| U113 - Kitchen garden | 58 |
| U120 - Forestry | 15 991 |
| U130 - Aquaculture and fishing | 1 |
| U140 - Mining and quarrying | 44 |
| U150 - Other primary production | 19 |
| U210 - Energy production | 13 |
| U221 - Manufacturing of food, beverages and tobacco products | 1 |
| U223 - Coal, oil and metal processing | 2 |
| U224 - Production of non-metal mineral goods | 3 |
| U225 - Chemical and allied industries and manufacturing | 1 |
| U226 - Machinery and equipment | 1 |
| U227 - Wood based products | 4 |
| U311 - Railway transport | 12 |
| U312 - Road transport | 164 |
| U313 - Water transport | 3 |
| U314 - Air transport | 12 |
| U315 - Transport via pipelines | 1 |
| U316 - Telecommunication | 2 |
| U317 - Logistics and storage | 17 |
| U318 - Protection infrastructures | 16 |
| U319 - Electricity, gas and thermal power distribution | 20 |
| U321 - Water supply and treatment | 8 |
| U322 - Waste treatment | 10 |
| U330 - Construction | 32 |
| U341 - Commerce | 38 |
| U342 - Financial, professional and information services | 2 |
| U350 - Community services | 99 |
| U361 - Amenities, museums, leisure | 326 |
| U362 - Sport | 167 |
| U370 - Residential | 378 |
| U411 - Abandoned industrial areas | 23 |
| U412 - Abandoned commercial areas | 2 |
| U413 - Abandoned transport areas | 4 |
| U414 - Abandoned residential areas | 30 |
| U415 - Other abandoned areas | 459 |
| U420 - Semi-natural and natural areas not in use | 5084 |
| **Total** | 58 428 |





*Author contributions.* All the authors processed and analyzed the data, wrote the paper, provided comments and suggestions on the manuscript.
B. E. and P. S. designed the survey methodology. B.E. and ESTAT are responsible of the LUCAS data collection.

*Competing interests.* The authors declare that they have no known competing financial interests or personal relationships that could have appeared to influence the work reported in this paper.