# Peer review of "LUCAS Copernicus 2018: Earth Observation relevant in-situ data on land cover and use throughout the European Union"

_Earth System Science Data, 2020_

## Referee Comment (RC1) · Radoux Julien (Referee) · 14 Sep 2020

General comments

This paper presents a unique dataset of major interest for the scientific community of EO in the EU. The protocol is rigourous which make me trust the quality of the dataset. I made however a few comments that could be worth to consider for the future collection.

Concerning the paper itself, I am not a native speaker so I cannot judge the language, but the complexity of the embedded datasets made it sometimes difficult to follow and some small changes in the structure would, in my opinion, make it more readable. For

the sake of completeness, I suggest to add the full description (including threshold that seprate "mixed" classes) of the land cover classes in the annex of the paper.

Specific comments

The title only mentions the land cover component, then the paper describes both land cover and land use attribute. The paper could focus on Land Cover only because it is already quite complex. Otherwise more details about Land use is necessary (e.g. what if the land use extent is not compatible with the land cover extent ?)

Line 49 (and after): it is not clear to me what is referred to by "EO limitation". From this paragraph, I was expecting limitation of the LUCAS dataset in order to be used by in EO workflow, but the three limitation are presented as typical shortcomings in "operationnal EO projects". I recommand to first focus on the reasons why in situ data in necessary for EO, then talk about the inherent shortcomings of a dataset designed to collect statistics (the latter is not explained IMHO) and that need to be addressed in the Copernicus module. Further details are given in section 3, but I think that this information is relevant in the introduction.

Line 85: so ∼1/3rd of the points of the grid have been surveyed in 2018 ? And they are selected according to a land cover based stratification ? Please clarify how you end up with 337854 points.

Line 95: How were the 90620 points selected ? random or stratified sampling ?

Line 120: what is meant by "exact location" of the observation ? If not defined by the 2km*2km grid, how is the point identified on the ground ? Is there a mark ? What is the precision of the geolocation (centimetric ? decimetric ? ). What is the precision of the distance measurement from the point.

How are the cardinal directions determined ? is it the geographic North, cartographic North or the magnetic North? For future work, I would suggest UTM north with a 45° shift (NE, NW, SE, SW) to be as close as possible to the standard Sentinel-2 grid (I see

on line 156 that LAEA is used to build the polygons, but for me the polygons should be created in UTM then projected in LAEA).

What is the minimum mapping unit of the distance estimate ? On figure 1 the polygon is obviously crop, but there are areas of bare soil in this crop field. My example is trivial, but what if there is a small shrub in a grassland or when is a gap in a forest considered as "not a tree". This needs to be specified in order to be used appropriately with respect to the spatial resolution of the EO data. The issue of heterogeneity at different spatial resolutions should be discussed at the end of the paper because it could have an impact on some studies.

Line 131: what if there are more than one land covers (depending on the direction)

Line 159: the homogeneity of the quadrilaterals is not guaranteed by the protocol. In the (unlikely) event that an item from another land cover is located inside the equilateral, is there a protocol to reduce the radial distance in one or two directions in order to exclude it ?

Line 164: do you mean that the point was unreachable ? With a 2 by 2 km grid, 11 km is really far away.

technical correction

Lines 1 and 20: Please replace "regular" by "evenly spaced" (or any more precise word), because regular could also be related to the temporal dimension.

Line 22: please clarify if there are 1351 293 unique geographic points point or if that number is the number of records (up to 5 records at each location)

Line 83: what is the purpose of the stratification if the sample is systematic ?

Line 96: 69.02 should be 69.92%

Line 97: I think that planned is better that programmed

Line 111 : double "with"

Line 177 : because of the orientation of the quadrilateral, I think that less than 50 pixels are fully included.

Line 178 : fo "subdecametric" sensors, the MMU becomes important.

Figure 4: what is the coordinate system of the map (Equirectangular ?) ? Please note that the orientation of the polygons is quite different from the cartographic North (LAEA effect ?)

---

## Referee Comment (RC2) · Radoux Julien (Referee) · 15 Sep 2020

In addition to my previous comments, I suggest to add the sampling probability of each point in an additionnal attribute field. This is indeed necessary for an appropriate use of the point in a statistically rigourous accuracy assessment.

---

## Referee Comment (RC3) · Ulf Mallast (Referee) · 26 Nov 2020

The manuscript by d'Andrimont et al. "summarizes the LUCAS Copernicus protocol to collect homogeneous land cover and proposes a methodology to create a ready-to-use dataset for Earth Observation land cover and land use applications with high resolution satellite imagery". Both aims are conducted in the frame of a new LUCAS module that is specifically tailored to EO, the LUCAS Copernicus module. The module but most of all a standardized way to provide in-situ data as training data for further LULC developments is of high importance and crucial. Both, the data set and the efforts of all persons involved are highly appreciated.

The present manuscript describes the protocol how in-situ data are obtained and proposes a methodology to create a ready to use dataset for further usage. It is well written, concise and informative. In certain parts it lacks clarity especially for an interested reader who is not a LULC expert. Some of it is induced by a diversity of used terms whose relation is not directly tangible. Some other relates to numerous points (LUCAS theoretical grid point, LUCAS Copernicus point point) and metrics (buffer and distances) for which it is unclear what really is of relevance and used during the survey or later on to e.g. define homogeneity. Regarding the term diversity a glossary may provide the essential light in the dark, while a simple process chart could disentangle the point/metric puzzle.

Since it is a data paper in which it is also asked to check the data quality and usability, let me mention that one decisive file was missing (LU-CAS_2018_Copernicus_attributes.csv). Thus I could not completely verify its content, but I will gladly do it as soon as the revised version of the manuscript is available.

These comments alongside further smaller ones are added to the *.pdf to facilitate preparing the final version of the manuscript that I would be happy to receive again.

Yet, overall the manuscript is very well developed and needs just some tweaks before being publishable. Congratulations.

Sincerly, Ulf Mallast

Please also note the supplement to this comment:
https://essd.copernicus.org/preprints/essd-2020-178/essd-2020-178-RC3-supplement.pdf

[revised manuscript text omitted]

---

## Author Comment (AC1) · 25 Jan 2021

We have answered point-by-point to all the comments raised by the referee in the attached PDF. The answers are provided below each comment in blue font. We would like to thank the reviewer for the quality and detailed reviewing provided. We would also like to notify the referee that this manuscript was selected as an Executable Research Compendia (ERC) pilot (see here https://o2r.info/pilots/ ) and the code to process the data and the table and figures will be available and reproducible after reviewing.

**Referee Comment 1**

General comments This paper presents a unique dataset of major interest for the scientific community of EO in the EU. The protocol is rigourous which make me trust the quality of the dataset. I made however a few comments that could be worth to consider for the future collection. Concerning the paper itself, I am not a native speaker so I cannot judge the language, but the complexity of the embedded datasets made it sometimes difficult to follow and some small changes in the structure would, in my opinion, make it more readable. For the sake of completeness, I suggest to add the full description (including threshold that seprate "mixed" classes) of the land cover classes in the annex of the paper.

**Specific comments**

The title only mentions the land cover component, then the paper describes both land cover and land use attribute. The paper could focus on Land Cover only because it is already quite complex. Otherwise more details about Land use is necessary (e.g. what if the land use extent is not compatible with the land cover extent ?)

This comment is valid and the title was modified to include "land use" especially as the land use extent is compatible with the land cover.

Line 49 (and after): it is not clear to me what is referred to by "EO limitation". From this paragraph, I was expecting limitation of the LUCAS dataset in order to be used by in EO workflow, but the three limitation are presented as typical shortcomings in "operationnal EO projects". I recommand to first focus on the reasons why in situ data in necessary for EO, then talk about the inherent shortcomings of a dataset designed to collect statistics (the latter is not explained IMHO) and that need to be addressed in the Copernicus module. Further details are given in section 3, but I think that this information is relevant in the introduction.

We thank the reviewer for this remark and have rephrased this section of the manuscript as the motivation of having a Copernicus module. The inherent limitation of the classical LUCAS for EO are described only in section 3.

Line 85: so ~1/3rd of the points of the grid have been surveyed in 2018 ? And they are selected according to a land cover based stratification ? Please clarify how you end up with 337854 points.

Indeed, the points were selected based on a land cover stratification. The stratification methodology is described in detail in "Redesign sample for Land Use/Cover Area frame

Survey (LUCAS) 2018"
(https://ec.europa.eu/eurostat/web/products-statistical-working-papers/-/KS-TC-18-006). This reference was missing and thus was added in the manuscript. We have also rephrased this section to increase readability.

About the resulting number of points (337854), it is also described in the document. To summarize, a sample can be defined as optimal both in terms of its costs (i.e. the number of units to be interviewed) and its accuracy (related to the sampling variance of target estimates) (Ballin, M. and Barcaroli, G., 2013). The approach followed in the optimisation process of LUCAS sampling design is based on the joint determination of the optimal stratification of a sampling frame, together with the optimal sample size determination and allocation. This approach is the most general one, as it can operate in the full multivariate case and its implementation is based on the use of the genetic algorithm.

*Ballin, M. and Barcaroli, G., 2013. Joint determination of optimal stratification and sample allocation using genetic algorithm. Survey Methodology, 39(2), pp.369-393.*

Line 95: How were the 90620 points selected ? random or stratified sampling ?

The sample of the Copernicus module was a third-phase sampling nested in the two-phase LUCAS core sampling scheme. This was clarified in the manuscript.

Line 120: what is meant by "exact location" of the observation ? If not defined by the 2km*2km grid, how is the point identified on the ground ? Is there a mark ? What is the precision of the geolocation (centimetric ? decimetric ? ). What is the precision of the distance measurement from the point.

The terminology "exact location" is not accurate. By this, we mean, the gps-measured location. The surveyor is aiming to reach the grid point but there is no mark on the ground. To locate it, the surveyor is using both a GPS and a ground document with the points displayed with an orthophoto in the background. The precision of the GPS varies according to devices, area and time but is recorded and provided along with the survey data. To avoid confusion, we have replaced the term "exact location" by "measured location".

How are the cardinal directions determined ? is it the geographic North, cartographic North or the magnetic North? For future work, I would suggest UTM north with a 45° shift (NE, NW, SE, SW) to be as close as possible to the standard Sentinel-2 grid (I see on line 156 that LAEA is used to build the polygons, but for me the polygons should be created in UTM then projected in LAEA).

A traditional magnetic compass has to be used to indicate a particular direction. North is indicated on both the topographic maps and the orthophotos. With the help of the compass, the surveyor can correctly orient the map and the orthophoto before examining them. Moreover, the compass is helpful to correctly take the photos in the cardinal directions (N, E, S, W).

We thank the reviewer for the suggestion. Indeed LAEA was used here and the authors will suggest to use UTM for the next survey.

What is the minimum mapping unit of the distance estimate ? On figure 1 the polygon is obviously crop, but there are areas of bare soil in this crop field. My example is trivial, but what if there is a small shrub in a grassland or when is a gap in a forest considered as "not a tree". This needs to be specified in order to be used appropriately with respect to the spatial resolution of the EO data. The issue of heterogeneity at different spatial resolutions should be discussed at the end of the paper because it could have an impact on some studies.

For the Copernicus module, one of the conditions to do the Copernicus survey is that " it is necessary that the extent of the Copernicus LC is at least 5m in any possible direction.". Therefore, the MMU of copernicus is 78.53 m² (i.e. a circle of 5 m radius).

For the core points, the LUCAS minimum mapping unit is about 7 m² ( a circle of 1.5 m radius) for points falling in a homogeneous area. When the land cover is not homogeneous, for example when it is composed of trees or shrubs interspersed with grass, the scale of observation has to be changed to classify it. In these cases a systematic observation of the "environment" in the vicinity of the point, which in LUCAS is called the extended window of observation, has to be adopted. The extended window of observation expands to a radius of 20 meters of distance (or 40 m diameter) from the point (representing an area of 0.13 ha).The window of observation has to be extended whenever the land cover at the point is heterogeneous. This occurs regularly in areas such as:
- permanent crops (B7X, B8X, except nurseries B83): parcels of permanent crops where the trees or other plants alternate with bare soils and/or grassland or another crop
- woodland (CXX)
- shrub land (DXX): where a mix of e.g. shrubs and trees might occur
- grassland (EXX), where land features may alternate (e.g. grassland with trees)
- bare land (FXX)
- wetland (HXX)

Line 131: what if there are more than one land covers (depending on the direction)

See the next question for an answer..

Line 159: the homogeneity of the quadrilaterals is not guaranteed by the protocol. In the (unlikely) event that an item from another land cover is located inside the equilateral, is there a protocol to reduce the radial distance in one or two directions in order to exclude it ?

Specifically for the Copernicus module,  the protocol reduces the radial distance is case of another land cover (note that Linear features with width < 3m are not considered a land cover change). The detailed field collection protocol of the Copernicus module including specific cases is available pp.57-67 of the C1 documents (https://ec.europa.eu/eurostat/documents/205002/8072634/LUCAS2018-C1-Instructions.pdf)

Line 164: do you mean that the point was unreachable ? With a 2 by 2 km grid, 11 km is really far away.

No, the aim of this filtering is to prevent potential encoding problems such as decimal or sign problems and thus remove macro error coming from encoding.

**technical correction**

Lines 1 and 20: Please replace "regular" by "evenly spaced" (or any more precise word), because regular could also be related to the temporal dimension.

Done.

Line 22: please clarify if there are 1351 293 unique geographic points point or if that number is the number of records (up to 5 records at each location)

The text was clarified by adding "at 651780 unique locations" indicating that the 1351 293 observations could be done at the same point for different years.

Line 83: what is the purpose of the stratification if the sample is systematic ?

The

Line 96: 69.02 should be 69.92%

Modified.

Line 97: I think that planned is better that programmed

Modified.

Line 111 : double "with"

Corrected.

Line 177 : because of the orientation of the quadrilateral, I think that less than 50 pixels are fully included.

The text was clarified "corresponding thus to almost 50 10-m pixels depending on the orientation".

Line 178 : fo "subdecametric" sensors, the MMU becomes important.

As explained in the previous MMU comment, they are 3 different MMU according to which data are selected:
- Copernicus module:  78.5 m² (a circle of 5 m radius)
- LUCAS points homogenous area :7 m² ( a circle of 1.5 m radius)
- LUCAS points homogenous area :1256.6 m² ( a circle of 20 m radius)

This was detailed in the manuscript.

Figure 4: what is the coordinate system of the map (Equirectangular ?) ? Please note that the orientation of the polygons is quite different from the cartographic North (LAEA effect ?)

The Fig is in EPSG:3857. Indeed, the polygon was created in LAEA and is then reprojected.

**Referee Comment 2**

In addition to my previous comments, I suggest to add the sampling probability of each point in an additionnal attribute field. This is indeed necessary for an appropriate use of the point in a statistically rigourous accuracy assessment.

This is a good suggestion and will be taken up for the LUCAS 2022 Copernicus campaign. The LUCAS Copernicus sampling was a third phase in the two phase LUCAS core sampling scheme. In addition, there was an additional requirement that the points sampled for the LUCAS grassland module (3500), and the LUCAS soil module (xxxxx), were also covered by the LUCAS Copernicus module. The stratification methodology is described in detail in "Redesign sample for Land Use/Cover Area frame Survey (LUCAS) 2018" (https://ec.europa.eu/eurostat/web/products-statistical-working-papers/-/KS-TC-18-006). The authors will pursue this in a forthcoming update.

---

## Author Comment (AC3) · 25 Jan 2021

We have answered point-by-point to all the comments raised by the referee in the attached PDF. The answers are provided below each comment in blue font. We would like to thank the reviewer for the quality and detailed reviewing provided. We would also like to notify the referee that this manuscript was selected as an Executable Research Compendia (ERC) pilot (see here https://o2r.info/pilots/ ) and the code to process the data and the table and figures will be available and reproducible after reviewing.

**Referee Comment 3**

The manuscript by d'Andrimont et al. "summarizes the LUCAS Copernicus protocol to collect homogeneous land cover and proposes a methodology to create a ready-to-use dataset for Earth Observation land cover and land use applications with high resolution satellite imagery". Both aims are conducted in the frame of a new LUCAS module that is specifically tailored to EO, the LUCAS Copernicus module. The module but most of all a standardized way to provide in-situ data as training data for further LULC developments is of high importance and crucial. Both, the data set and the efforts of all persons involved are highly appreciated.

The present manuscript describes the protocol how in-situ data are obtained and proposes a methodology to create a ready to use dataset for further usage. It is well written, concise and informative. In certain parts it lacks clarity especially for an interested reader who is not a LULC expert. Some of it is induced by a diversity of used terms whose relation is not directly tangible. Some other relates to numerous points (LUCAS theoretical grid point, LUCAS Copernicus point point) and metrics (buffer and distances) for which it is unclear what really is of relevance and used during the survey or later on to e.g. define homogeneity. Regarding the term diversity a glossary may provide the essential light in the dark, while a simple process chart could disentangle the point/metric puzzle.

Thank you for the suggestions provided. We have added a glossary to the manuscript. We have also tried to clarify the whole text of the manuscript. However, we haven't found how to figure out the data with an additional processing chart as the data combines survey and processing. However, we believe that Figure 1 is a good illustration to clarify and have thus tried to improve the related text for the sake of clarity.

Since it is a data paper in which it is also asked to check the data quality and usability, let me mention that one decisive file was missing (LUCAS_2018_Copernicus_attributes.csv). Thus I could not completely verify its content, but I will gladly do it as soon as the revised version of the manuscript is available.

The file should have been available previously. After updating the files, we have also uploaded them to the JRC open data catalog : http://data.europa.eu/89h/cfe66a0c-bdee-4074-96e1-a2f7030b9515.

These comments alongside further smaller ones are added to the *.pdf to facilitate preparing the final version of the manuscript that I would be happy to receive again.

Yet, overall the manuscript is very well developed and needs just some tweaks before being publishable. Congratulations.

Sincerly, Ulf Mallast

We thank the reviewer for the useful comments and suggestions. We have tried to include them all and are addressing them point by point below.

**Comments in the PDF**

L71: Is the copernicus data the same as the LUCAS Copernicus module? and the LUCAS Copernicus in-situ survey? and LUCAS core?

At certain locations, and coming from a different field compared to the authors, it is sometimes hard to follow the narrative, given the diversity of terms that is not really clear. I suggest two approaches: 1. I would encourage the authors to include a gloassary and 2. possibly even a process chart on how each of the terms are interrelated (if the latter makes sense).

Thank you for the suggestions  provided. We have added a glossary at the end of the manuscript. We have also tried to clarify the whole text of the manuscript. However, we haven't found how to figure out the data with an additional  processing chart as the data combines survey and processing.

L89: A question that directly pops up in my head, when it comes to design is the following: Are the LUCAS points always the same for each of the survey years?

The 2-km grid is static and contains about 1.1 million of points. Out of the 337,854 points surveyed in 2018, 23% points had been included in three previous surveys (2009, 2012, and 2015), 25% had already been surveyed once or twice before (e.g. in 2009 and 2015), and the remaining 52% of the points were new entries.

And, unaware of the exact locations of each of the survey points, it would make a lot of sense to include long-term observatories/sites (see here: DEIMS.org) that record their land use/cover in regular intervals (in in the near future also in a standardized fashion) but spatially explicit. In turn, one would not only rely on a regular LUCAS grid every 2km but could include spatially continuous data that in itself contains gradients that the grid cannot depict.

Indeed, it would be interesting to link with ecologically monitored zones such as DEIMS. However, the goal of LUCAS is to collect statistics about Land Use and Land Cover along with their dynamic though time at the EU level. The sampling scheme and methodology is thus designed for that and cannot be modified.

L92: As idea for the future, it might also be worthwile to include RTK drones data that are amazingly cheap and provide a high resolution NADIR view of the LUCAS polygon as well. The latter could be evaluated concerning homogeneity or vegetation gaps/density.

This would indeed be very valuable information however there is currently no plan to use drones for LUCAS 2022 campaign. This is expensive and need to have a policy support of the different involved DGs of the commission. For your information, the EMBAL (European Monitoring of Biodiversity in Agricultural Landscapes) project which is currently in a pre-pilot phase is looking at this option. However, because this type of survey is pretty expensive at the EU-level and different national uav regulations are in place, there is , to the best of our knowledge, no plan for such a survey.

L106-109: If I understood correctly, the observer assigns the land cover from the observation point to the unreachable LUCAS point.

If so, in cases of a LUCAS point being located at or in close proximity to land cover boundary , it may happen that a LUCAS point is assigned a wrong land cover. Can this be the case and is the observer asked to note down (e.g. in the metadata) the actual obervation point?

The surveyor has a GPS and ground document with a map of the points on a very high resolution orhophoto to help him to locate the point he has to reach. If he cannot reach it, he registers an information in the metadata ("obs_type": 1 - In situ < 100 m; 2 - In situ > 100 m ... ) along with the distance to the point ("obs_dist"). For the Copernicus point, the distance to the point should be less than 100 m. In case he cannot reach the point, he is asked to note down the actual observation point Land cover at level 2 in addition to the one observed on the LUCAS grid points.

L116: "fraction" is enough, I would even suggest to think about providing the exact percentage (1-7%), given the fact that fraction is very descriptive

"Subfraction" was changed to "fraction" and the percentage was added.

L119: What is the accuracy? Are surveyors requested to use a high end GPS device with submeter accuracy or even subcentimeter accuracy or is it sufficient to use GPS integratd in mobile phones? The later is in the range of 5-10m and would certainly pose a risk.

The hand-held GPS receiver has not a sub-meter accuracy which could indeed pose a risk of location inaccuracy. To limit the risk, the surveyor uses a ground document includes a topographic map (the most widely used types of maps with a scale usually between 1:10.000 and 1:200.000) and an orthophoto (an orthorectified aerial photograph, thatis, free from the distortion caused by inclination angle and relief, with a scale that normally, varies between 1:10.000 and 1:2.000). The ground document is the base for the surveyor to locate the point and to estimate the area of the parcel the point is located in. Additionally the precision of the GPS is collected (and available as metadata) and the location of the points are cross checked afterwards in the office.

L121-122: But how is it done in the field? I assume the surveyor estimates it visually from her/his earth-bound observation point, and this is very subjective and error prone.
Here especially I would suggest to include drones to either get a better estimation or even provide a "homogenity-factor".

To measure the distance, the surveyor uses the GPS and the ground document on which the grid points  are displayed with the orthophoto in the background. If needed, measurement could be done also a posteriori in the office using the orthophoto. The detailed field collection protocol of the Copernicus module including specific cases is available pp.57-67 of the C1 document (https://ec.europa.eu/eurostat/documents/205002/8072634/LUCAS2018-C1-Instructions.pdf) Using drones would indeed be optimal but, as described previously, it is not planned for the next campaign.

L126: How do the 50m relate to the radii of 1.5 and 20m mentioned before. Is it just another feature the surveyor has to record?

The  question leads me back to the general sampling design of sampling point spacing of 2km. Why not look and predefine a homogenous area in close proximity to LUCAS points with the 2km spacing instead of insisting on these 2km LUCAS points? That way heterogeneity would not play a role.

And second, if the surveyors are provided a bird-view map from these points (from drones or VHR sateliites e.g. Quickbird), with exact distances and angles to landmarks the geolocation uncertantiy may be reuduced. Even more so, if certain marks can be permantently installed (such as the plastic head plate fom surveyors) that define the point of interest.

For the first part of the question, the general sampling design is a stratified sample (n=337,854)  selected among a systematic 2km grid (n=1.1 million).

On these points the observation radius is **1.5m** if the point is falling in a homogeneous area.

When the land cover is not homogeneous, for example when it is composed of trees or shrubs interspersed with grass, the scale of observation has to be changed to classify it. In these cases a systematic observation of the "environment" in the vicinity of the point, which in LUCAS is called the extended window of observation, has to be adopted. The extended window of observation expands to a radius of **20 meters** of distance.

Finally, on a subset of the points which are part of the Copernicus module, an additional extent information is collected up to **50 m**.

About the second remark, this is a valid remark and indeed, it would be more convenient to have a very high resolution image (from a drone or a satellite) concurrent to the survey. However, from a practical point of view, it is difficult to obtain and much more expensive. For the CAP monitoring the member states are usually collecting orthophoto with a submetric resolution at least once every 3 years and the data could then be re-used for this exercise.

However, we fully agree with the reviewer that having such type of data would be very valuable.

L154: Again, I am a bit confused, how do the Copernicus Point and the theoretical grid point relate to each other? I thought that if the surveyor could reach the theoretical point the theoretical grid points is equal to the Copernicus point. But if only 1.5 are evaluated in terms of homogeneity, how can the entire polygon represent a homogenuos area?

To disentangle the puzzle, I suggest to include a simple processing chart that pictures what the authors have described in the text..

The Figure 1 aims to clarify the puzzle. Indeed, if the surveyor could reach the theoretical point, the LUCAS copernicus point and the LUCAS grid point should be the same. The Copernicus polygon is homogenous.

L157-159: I do not understand a) how the number is generated and b) why it is important. Can the authors please elaborate?

The Table 1 describes in detail the numbers. Out of the 337,854 points surveyed in 2018, 63,364 were effectively also surveyed with the Copernicus protocol. The number is important as it is the total number of points for which the Copernicus protocol was applied. To clarify, we have added a reference to Table 1.

L165: the conversion is very general and with these numbers only valid at the equator. Given that Europe is the spatial frame for the current MS, I suggest to provide a better conversion.

Indeed, it was not realistic for the longitude. We have modified the manuscript calculating the metric distance based on a latitude of 50 degree N.

L169: sometimes it is "polygon" sometimes it is "point". Please be consistent.

Indeed, thanks for the remark, we have modified accordingly to be consistent.

L174: 2?

Indeed, it was a typo, thanks.

FIGURE3 : If I may suggest, it would round up the overall impresssion, if the geospatial graphics have the same layout and design. Could the authors please adapt this figure to figure 1 regarding e.g. the frame, the coordiantes, axis titles etc.

The Figure 3 was generated by a R script while Figure 1 was done with QGIS, thus it is impossible to have the same layout. Also, as this manuscript is part of a pilot to generate an ERC (Executable Research Compendium) with reproducible code (https://o2r.info/), we prefer to have different layout but reproducible code.

Since level-2 data were mentioned before, I would naturally assume to see level-2 data in the graphic. If possible and still comprehensible, I would thus suggest to present level-2 here, instead of level-1, if the authors find it presentable with 26 classes.

Indeed, we have tried with the level-6 26 Classes but it was not presentable.

L179: I have mentioned it before, for the sake of clarity, it would be worthwile thinking about a glossary/ scheme to assign more detail to these terms (LUCAS core data, Copernicus polygons, Copericus module)

Along with the Figure 1, a glossary has been added a t the end of the manuscript describing the following terminology:
LUCAS core data,
LUCAS theoretical grid point,
LUCAS Copernicus point
LUCAS Copernicus polygons,
LUCAS Copernicus module

L213-214: I mentioned with the integration of long-term monitoring sites (observatories) and usage of drones in future surveys two points before, that could also be mentioned here and shorty discussed, as it would represent an interesting option and provide further added value.

Because of LUCAS main objective and methodology, i.e. collecting statistics about Land Use and Land Cover systematically over the whole EU, the integration of long-term monitoring sites seems unlikely to happen within this survey framework. About the drone survey, we have added a comment in the discussion of the manuscript.

L217: wouldn't it be relevant to all sensors in the high resolution section (=<30m GSD)? It is intended for 10m Sentinel due to Copernicus affiliation, but genuinely it could be used for Landsat, Quickbird, ASTER, WorldView etc.

This was added in the manuscript.

L226: If I am not mistaken the decisive csv file with all attributes is missing. Can the authors please check?

The file should have been available previously. After updating the files, we have also uploaded them to the JRC open data catalog.